# Design of a Memristor-Based Digital to Analog Converter (DAC)

**Ghazal A. Fahmy *** and **Mohamed Zorkany**

Electronics Department, National Telecommunication Institute, Cairo 11768, Egypt; m_zorkany@nti.sci.eg
* Correspondence: ghazal.fahmy@nti.sci.eg

**Abstract:** A memristor element has been highlighted in recent years and has been applied to several applications. In this work, a memristor-based digital to analog converter (DAC) was proposed due to the fact that a memristor has low area, low power, and a low threshold voltage. The proposed memristor DAC depends on the basic DAC cell, consisting of two memristors connected in opposite directions. This basic DAC cell was used to build and simulate both a 4 bit and an 8 bit DAC. Moreover, a sneak path issue was illustrated and its solution was provided. The proposed design reduced the area by 40%. The 8 bit memristor DAC has been designed and used in a successive approximation register analog to digital converter (SAR-ADC) instead of in a capacitor DAC (which would require a large area and consume more switching power). The SAR-ADC with a memristor-based DAC achieves a signal to noise and distortion ratio (SNDR) of 49.3 dB and a spurious free dynamic range (SFDR) of 61 dB with a power supply of 1.2 V and a consumption of 21 μW. The figure of merit (FoM) of the proposed SAR-ADC is 87.9 fj/Conv.-step. The proposed designs were simulated with optimized parameters using a voltage threshold adaptive memristor (VTEAM) model.

**Keywords:** memristor; DAC; ADC; sneak path

## 1. Introduction

Chua has discovered the existence of a passive two-terminal fundamental element, which is called a memristor [1]. This element is characterized by the relationship between charge flux–linkage $\phi(t)$ and charge $q(t)$, where $\phi(t) = \int_{-\infty}^{t} v(t)dt$ and $q(t) = \int_{-\infty}^{t} i(t)dt$ respectively. This element has been presented as a fourth fundamental element. However, since that time, no physical element with $f(\phi, q) = 0$ characteristics had been yet discovered. Finally, in 2008, the Hewlett–Packard (HP) team proved the practical existence of the memristor [2]. Since 2008, the memristor has become an interesting research topic and attracted many scientists to conduct research on memristors in terms of several fields (such as modeling, fabrication, and characterization) and employ it towards different applications.

The change of flux–linkage $\phi(t)$ due to the change of charge $q(t)$ is known as memristive change (M), expressed as $Wb/c$ or $\Omega$, where $M = d\phi/dq$. The memristor has a characteristic behavior proposed by Chua [1] and modeled by at least 15 transistors and some passive components to emulate the behavior of a single memristor. Figure 1 shows the relationship between the basic two-terminal elements: resistor, capacitor, inductor, and memristor.

Nowadays, memristor technology has entered different and important domains. It can be used to build neuromorphic biological systems, artificial neural networks (ANN) (based on its good ability to retain digital memory), and logic circuits. These features also enable it to play a role in new digital computer technology. It is very efficient to use in smart remote sensing and Internet of Thing (IoT) applications, as it has a very low power consumption. Due to these features of memristors, they have an efficient and effective way to store digital data (as well as analog data) with power-efficient criteria [3].

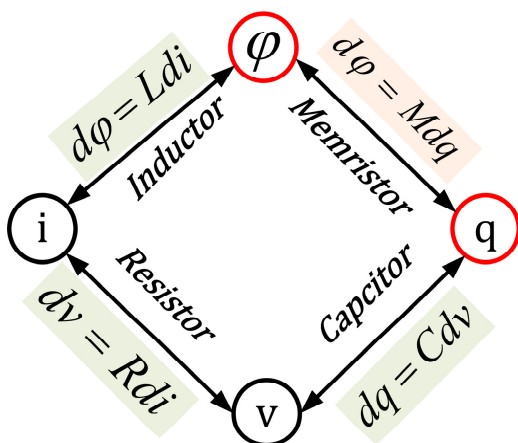

**Figure 1.** The relationship between the basic two-terminal elements (resistor, capacitor, inductor, and memristor).

The memristor has been used in non-linear analog circuit designs (such as oscillators and digital to analog converters), since it has a lower area and is faster than the conventional CMOS circuit. The memristor is a natural counterpart to the fundamental digital to analog relationship, as the memristance value changes according to the applied voltage. Along with this, the memristor's current state depends on its previous state, so it was recently used to design a digital to analog converter (DAC). Since the DAC is the main component of all data-driven acquisition circuits, it is the link between the analog signals of the real world and the digital signal processing field. So much of the applied research is conducted to overcome the design challenges of DAC circuits, such as its low area and low power [4]. One of the newest ways to overcome these challenges is to use memristor technology [5]. Therefore, there are a few different research directions for the use of memristors in DAC designs. For example, L. Danial et al. were able to calibrate a DAC design configuration using neural networks as an artificial intelligence technique [6]. Another kind of research implemented hybrid DAC and ADC with a hybrid circuit using memristors. As L. Gao et al. demonstrated, DAC and Hopfield-network ADC circuits are able to utilize post-fabrication resistance tuning features [7]. Additionally, F. Cai et al. designed an integrated CMOS re-programmable memristor system based on DAC and ADC for the efficient multiplication of accumulated operations, and they subsequently used that design to implement a multi-layer neural network [8].

A 2 bit hybrid memristor and a CMOS technology-based DAC were presented in [9], using a non-overlapped signal to achieve the basic cell for DAC. In this model, the DAC area does not significantly reduce due to the use of five transistors, one resistor, and a buffer circuit, all of which affect the total area of DAC. Moreover, the author did not declare how to extend his proposed 2 bit DAC to n bit DAC. In this work, we proposed a 2 bit memristor DAC without an included resistor element, transistors, or buffer that will reduce the DAC area significantly. Moreover, the n bit DAC has been addressed in this work and has been verified by two case studies. There are several designs using a memristor to develop an analog to digital converter (ADC) [10,11]. For example, a hybrid CMOS–memristor logic gate (known as memristor-ratioed logic (MRL)) has been deployed in the digital blocks of the pipeline ADC to decrease the clock delay further and increase the logic density [12,13]. A memristor-based flash ADC was also provided in [14]. In this work, a successive approximation register analog to digital converter (SAR-ADC) using the proposed memristor DAC has been provided to achieve low area and low power, as will be explained in Section 5.

In this work, a basic cell for DAC that uses a memristor element has been proposed. The proposed DAC cell was used to design a thermometer code to analog and study the characteristics of DAC. The binary-weighted DAC cell based on two memristors was proposed with a sneak path solution.

The design of memristor-based circuits still faces some challenges and problems. One of the main challenges at the architecture and circuit level is the sneak path current problem [15]. Some solutions to this problem established a sophisticated classification, like the one diode–one memristor model (1D1M), the one transistor–one memristor model (1T1M), and the one selector–one memristor model (1S1M). Some other solutions depend on self-rectifying and self-selective memristors [16]. More research is still urgently needed to overcome this problem.

This paper is organized as follows: Section 2 presents the memristor element characteristics. Section 3 focuses on the ratioed logic (MRL logic) of how the circuit operates. Section 4 proposes a DAC cell based on memristors, with accompanying simulations. There are four subsections that cover the thermometer DAC and binary DAC, with a case study conducted on each of them. Section 4.1 includes the proposed 2 bit DAC cell and its simulation, while Section 4.2 explains how to use the proposed DAC cell to design a thermometer code to analog conversion. Section 4.3 explains the proposed binary-weighted memristor DAC cell and then explains how to extend it to n bit DAC, while Section 4.4 studies the sneak path issues and employs one of the derived solutions in the proposed design. Section 5 covers the case study for the memristor DAC and its use in the SAR-ADC by replacing the conventional capacitor DAC with the proposed memristor DAC. The last section explains the conclusion of this work, followed by the references.

## 2. Memristive Element Characteristics

There are numerous mathematical models developed by researchers to describe the behavior of memristors. The linear ion drift model, the non-linear ion drift model, the Simmons tunnel barrier model, the threshold adaptive memristor (TEAM) model, Yakopcic's model, and the voltage threshold adaptive memristor (VTEAM) are some of the widely accepted models in the literature. The VTEAM model is used in this work due to its ability to fit most of the switching characteristics exhibited by memristor devices. Moreover, it is an accurate model to represent non-linear memristor behaviors. In order to study the characteristics of the memristor element of the VTEAM model, the relationship between voltage and current in the memristor element is expressed in the following mathematical model [9]:

$$v(t) = [R_{on}\frac{W(t)}{D} + R_{off}(1 - \frac{W(t)}{D})]i(t) \tag{1}$$

$$\frac{dw(t)}{dt} = \begin{cases} k_{off}(\frac{v(t)}{V_{off}} - 1)^{\alpha_{off}} f_{off}(w), 0 < V_{off} < v \\ 0, V_{on} < v < V_{off} \\ k_{on}(\frac{v(t)}{V_{on}} - 1)^{\alpha_{on}} f_{on}(w), v < V_{on} < 0 \end{cases} \tag{2}$$

where $v(t)$ is the voltage across the memristor terminals, $i(t)$ is the current passing through the memristor, and $\alpha_{off}$, $\alpha_{on}$, $k_{off}$, and $k_{on}$ are fitting parameters. $V_{on}$ and $V_{off}$ are threshold voltages. $f_{on}(w)$ and $f_{off}(w)$ are window functions used to limit w to have a value between 0 and 1. This state variable represents the state of the memristor.

In Equation (1), $v(t)$ can be written as time varying resistance ($R_m(t)$), which can multiply the memristor by $i(t)$ as follows:

$$v(t) = R_m(t)i(t) \tag{3}$$

where $R_m(t)$ will be written as

$$R_m(t) = R_{on}\frac{W(t)}{D} + R_{off}(1 - \frac{W(t)}{D}) \tag{4}$$

where $R_{on}$ is on-state resistance (with a high dopant concentration) and $R_{off}$ is off-state resistance (with a low dopant concentration). A non-linear relationship between voltage and current in Equation (1) results in the relationship exhibiting pinched hysteresis loop

I–V characteristics. Several models have been proposed for memristors, some of which did not include a threshold voltage or a current parameter. Other models have included the threshold current parameters (such as the TEAM model). However, the experiment proved the existence of threshold voltage. The VTEAM Model has included the threshold voltage that satisfied these experimental results and has presented sufficient accuracy compared to that of the existing model. Figure 2 shows the ideal memristor I–V curve with two different setups. The difference between the setup circuits of Figure 2a,b is the flipping of the memristor direction. Consequently, the I–V curve illustrates the opposite state of a memristor. In setup (a), the memristor element changes from off-state to on-state when the applied voltage is greater than $+V_{th}$. However, in setup (b), the memristor element changes from off-state to on-state when the applied voltage is less than $-V_{th}$.

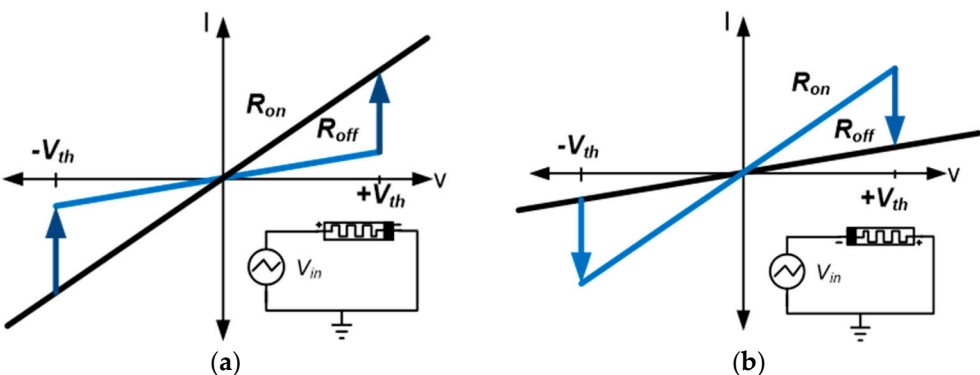

(**a**)                                   (**b**)

**Figure 2.** The ideal curve of current–voltage characteristics when (**a**) the positive terminal is connected to the supply and the negative terminal is connected to the ground and (**b**) the negative terminal is connected to the supply and the positive terminal is connected to the ground.

*Memristor Setup and Characteristic*

The VTEAM model has been employed to test the characteristics of the memristor device [17]. Figure 3 shows the characteristic setup circuit. The VTEAM model's parameters included in this setup are $V_{on} = -0.2$ V, $V_{off} = 0.02$ V, $R_{on} = 50 \, \Omega$ (for on-resistance), and $R_{off} = 1000 \, \Omega$ (for off-resistance), with delta-time ($dt$) set to be $10^{-12}$ for simulation purposes and the input signal is a sine-wave with a frequency of 1 MHz and a peak voltage of 1 V. A cadence tool has been used to simulate the VTEAM verilog-A model. The VTEAM model parameters used in the setup simulation are as follows in Table 1.

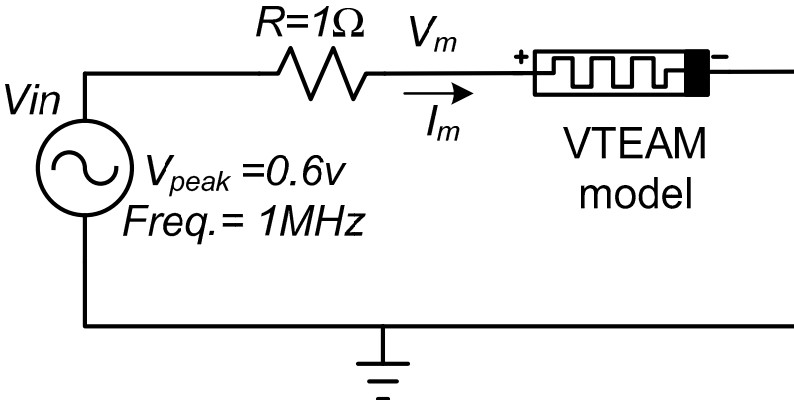

**Figure 3.** Setup circuit for the simulation memristor voltage threshold adaptive memristor (VTEAM) model.

**Table 1.** The VTEAM model parameter used in setup circuit for simulation.

| $a_{off}$ | 3 | $R_{off}$ [$\Omega$] | 1000 |
|---|---|---|---|
| $a_{on}$ | 1 | $R_{on}$ [$\Omega$] | 50 |
| $V_{on}$ [V] | $-0.2$ | $D$ [nm] | $3 \times 10^{-9}$ |
| $V_{off}$ [V] | 0.02 | $w_{off}$ [nm] | $3 \times 10^{-9}$ |
| $k_{off}$ [m/s] | $5 \times 10^{-4}$ | $w_{on}$ [nm] | 0 |
| $k_{on}$ [m/s] | $-10$ | $w_{init}$ [nm] | 0 |

Figure 4 shows the characteristics of the memristor device with the previous parameters instated. The I–V relation curve illustrates the pinched-off circuit at the zero crossing and has voltage thresholds at $V_{th1} = -0.2$ V and $V_{th2} = 0.3$ V, which prove that the memristor device has two-threshold voltages, as presented in [17].

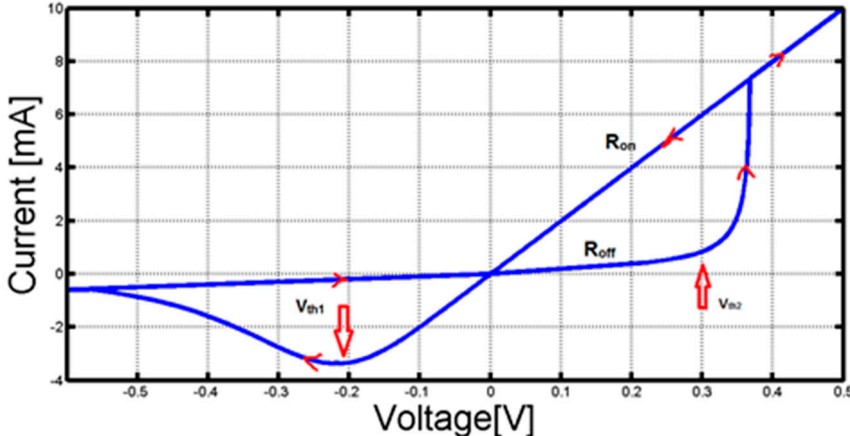

**Figure 4.** The I–V simulation curve of setup 3.

## 3. Memristor Ratioed Logic (MRL)

Many approaches have been proposed to implement logic gate circuits employing only memristor elements in [18,19], while in other approaches, hybrid memristor elements and CMOS structures are used to implement logic gates [20]. Basic logic gates (such as AND- and OR-gates) have been provided using two memristors combined with the appropriate polarities [20]. This approach has been simulated and illustrated in this work to deeply understand the connection between two memristors and how it works as follows: Figure 5a shows an AND-gate where the $V_{in1}$ and $V_{in2}$ are connected to a negative memristor sign and the positive memristor sign it is connected to gathers the output voltage $V_{o1}$. The $V_{o1}$ is expressed as follows:

$$V_{o1} = \frac{R_{m1} * V_{in2} + R_{m2} * V_{in1}}{R_{m1} + R_{m2}} \tag{5}$$

where $R_{m1}$ and $R_{m2}$ are the memristor resistances (which will be on-resistance ($R_{on}$) or off-resistance ($R_{off}$) according to the applied voltage on the corresponding memristor). Table 2 illustrates the different states of each memristor according to the applied voltage. This topology functions as an AND-gate. Figure 5b shows an OR-gate where $V_{in1}$ and $V_{in2}$ are applied to a positive memristor sign and the negative memristor sign it is connected to gathers the output voltage $V_{o1}$ and is expressed in Equation (5). Table 3 explains the functionality of the OR-Gate and the status of each memristor element.

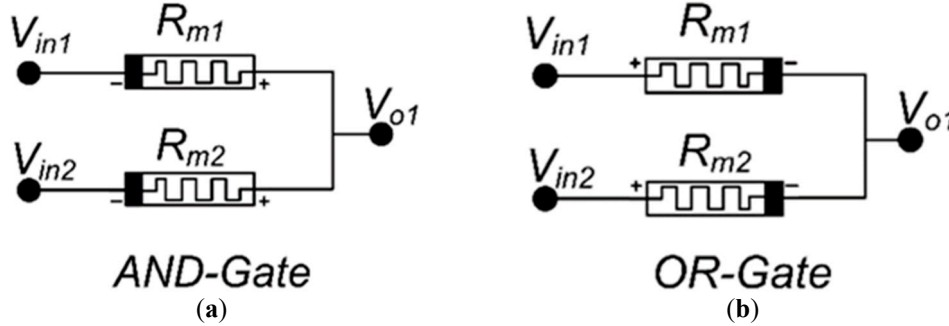

**Figure 5.** Two memristors connected as (**a**) an AND-gate and (**b**) an OR-gate.

**Table 2.** The different states of each memristor for Figure 5a.

| $V_{in1}$ | $V_{in2}$ | $R_{m2}$ | $R_{m1}$ | $V_{o1}$ |
|-----------|-----------|----------|----------|----------|
| High | High | $R_{off}$ | $R_{off}$ | High |
| High | Low | $R_{on}$ | $R_{off}$ | Low |
| Low | High | $R_{off}$ | $R_{on}$ | Low |
| Low | Low | $R_{off}$ | $R_{off}$ | Low |

**Table 3.** The different states of each memristor for Figure 5b.

| $V_{in1}$ | $V_{in2}$ | $R_{m2}$ | $R_{m1}$ | $V_{o1}$ |
|-----------|-----------|----------|----------|----------|
| High | High | $R_{on}$ | $R_{on}$ | High |
| High | Low | $R_{off}$ | $R_{on}$ | High |
| Low | High | $R_{on}$ | $R_{off}$ | High |
| Low | Low | $R_{off}$ | $R_{off}$ | Low |

## 4. Proposed DAC Cell Based on Memristor Technology

The most two common types of DAC architecture are a thermometer DAC and a binary-weighted DAC. In this work, a basic cell for DAC architecture will be proposed based on memristor technology for both types of DAC. Moreover, two case studies using the proposed DAC architecture will be provided. The first case study concerns a thermometer code to analog converter-based 2 bit memristor-based DAC. However, the second case study concerns a memristor DAC-based successive approximation register (SAR-ADC). As the design of memristor-based circuits faces sneak path current problems, we will propose a solution to overcome this problem based on a low power amplifier. The following subsections explain the proposal in detail.

### 4.1. DAC Cell Based on a Memristor Device

The basic structure of a thermometer-coded DAC has an equal resistor (current–source segment) for each output value of DAC. Designing the thermometer DAC cell using memristor technology will add all of the advantages of memristors to DAC, such as a good efficient way to store digital and analog data with power-efficient criteria. The proposed thermometer DAC cell depends on the use of two memristors connected with an opposite sign. Figure 6a,b show a basic DAC cell using a memristor, where the two memristors are structured in opposite signs. The $V_{in1}$ is applied on the negative side of $R_{m1}$ and the $V_{in2}$ is applied on the positive side of $R_{m2}$. The positive and negative sides of $R_{m1}$ and $R_{m2}$, respectively, are connected to the output node $V_{o1}$. $R_m$ could be on-state or off-state, according to the direction of the memristor at certain values of $V_{th}$. The two memristors have the same value for off-resistor ($R_{off} = 1000\ \Omega$) and on-resistor ($R_{on} = 50\ \Omega$). In this work, the input voltage will be 0 V or 1 V. When the applied voltage $V_{in1}$ is 1 V and $V_{in2}$ is

0 V, the current will flow from $V_{in1}$ to $V_{in2}$. Consequently, $R_{m1}$ and $R_{m2}$ will change to $R_{off}$, and the $V_{o1}$ will be 0.5 V. On the other hand, when the applied voltage $V_{in2}$ is 1 V and $V_{in1}$ is 0 V, the current will flow from $V_{in2}$ to $V_{in1}$. Consequently, $R_{m1}$ and $R_{m2}$ will change to $R_{on}$, and the $V_{o1}$ will be 0.5 V as well. In these two cases, the output voltage is the same but the current of the second case is much greater than that of the first case due to $R_{on}$ being much less than $R_{off}$. Figure 7 shows the simulation of the third topology with the following sequence of inputs for ($V_{in1}$,$V_{in2}$): (0,0), (0,1), (1,1), (0,1), and (0,0). The output voltage and memristor current are shown in Figure 7. The output voltage will be three levels (according to this sequence of inputs), while the current will be either 0 or 10 mA. If the sequence of inputs has been replaced with (0,0), (1,0), (1,1), (1,0), and (0,0), the output voltage will be the same while the current will be reduced significantly due to entering the off-state of the memristor, which has high resistance ($R_{off}$) as shown in Figure 8. This topology will function as the basic cell for structuring the digital to analog converter to convert the thermometer code to an analog signal.

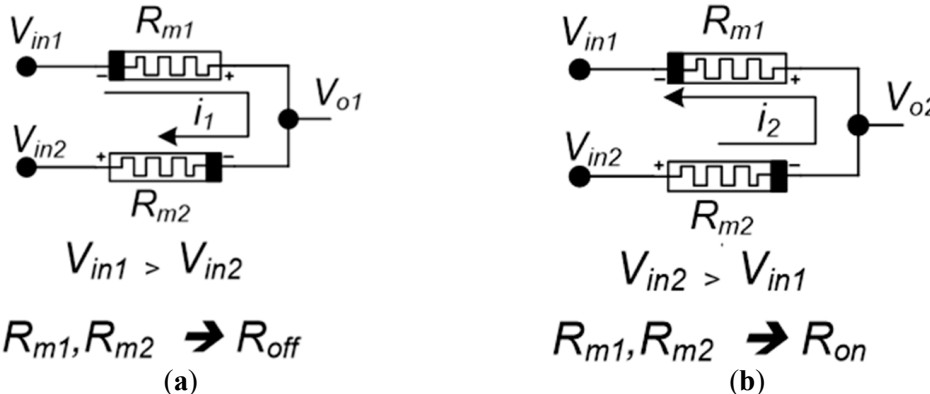

**Figure 6.** The proposed digital to analog converter (DAC) cell, with two memristors connected with an opposite sign. (**a**) When $V_{in1} > V_{in2}$. (**b**) When $V_{in2} > V_{in1}$.

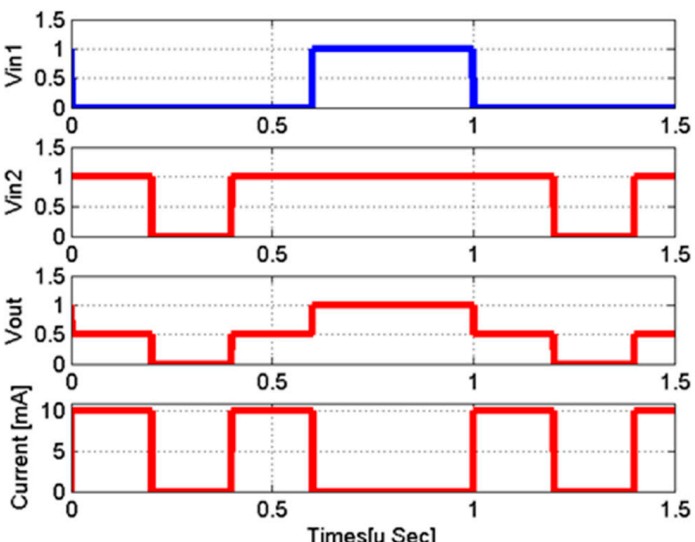

**Figure 7.** Simulation of the proposed DAC cell with the following sequence of inputs for ($V_{in1}$,$V_{in2}$): (0,0), (0,1), (1,1), (0,1), and (0,0).

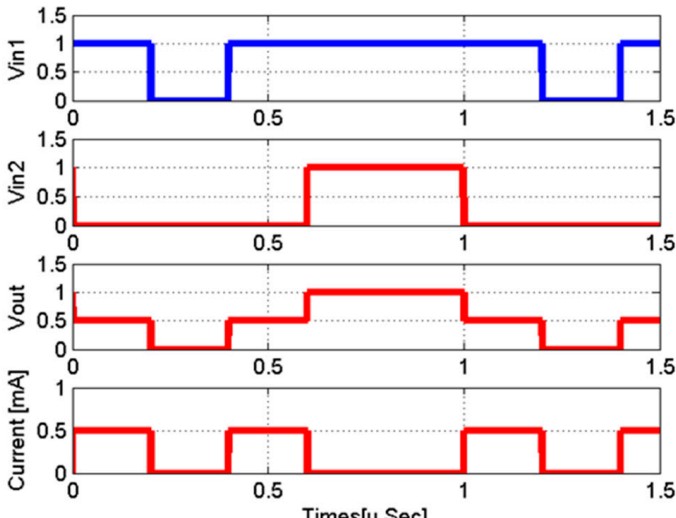

**Figure 8.** Simulation of the proposed DAC cell with the following sequence of inputs for ($V_{in1}, V_{in2}$): (0,0), (1,0), (1,1), (1,0), and (0,0).

### 4.2. Design of the Thermometer Code to Analog Converter Based on Memristor

A thermometer code to analog signal is a type of digital to analog converter (DAC). It is required to convert a thermometer code to an analog signal. It can use the conventional resistance string or a capacitor array, which requires much more area to implement. By using the memristor, however, it can be realized in a small area with low power consumption.

An eight level thermometer code was used to construct the memristor DAC, as shown in Figure 9, which contains three stages. Stage one is an input stage for the binary data $D_0$, $D_7$, while stage three is the output stage, containing one DAC cell. Each DAC cell includes two connected memristors in the opposite sign. In order to extend the DAC input from 8 to 16, the input of the DAC cell should double. The following equation describes the total number of DAC cells required for n number of inputs:

$$C_t = b_n - 1 \tag{6}$$

where $b_n$ is the number of inputs and $C_t$ is the total number of DAC cells required to achieve the DAC circuit. The total amount of memristors required for DAC is $2(b_n - 1)$. Figure 10 shows the simulation for the DAC circuit in Figure 11.

The integral non-linearity (INL) is one of the DAC specifications used to measure the deviation between the ideal output value and the actual output value. This deviation should be less than $\pm 0.5$ least significant bit (LSB). The LSB value is 100 mV in this case. The INL simulation has been illustrated in Figure 11. It demonstrates that INL is 8 mV when time is greater than 1 µs, but at the starting time, the maximum INL value of 33 mV was recorded. The INL achieved less than $\pm 0.5$ LSB.

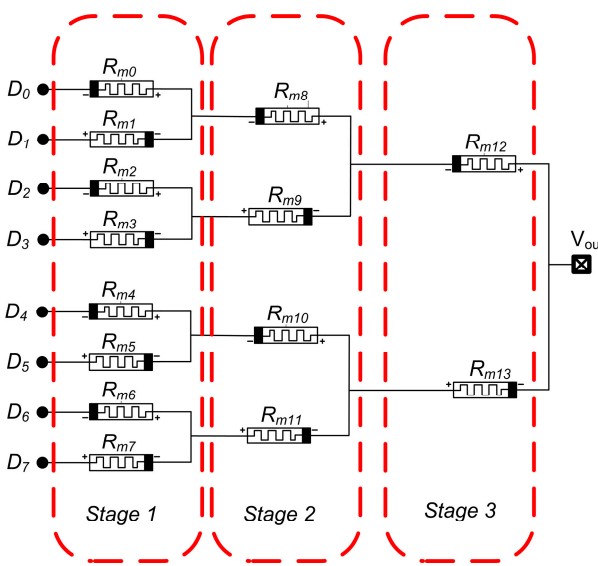

**Figure 9.** An eight level input memristor-based DAC circuit.

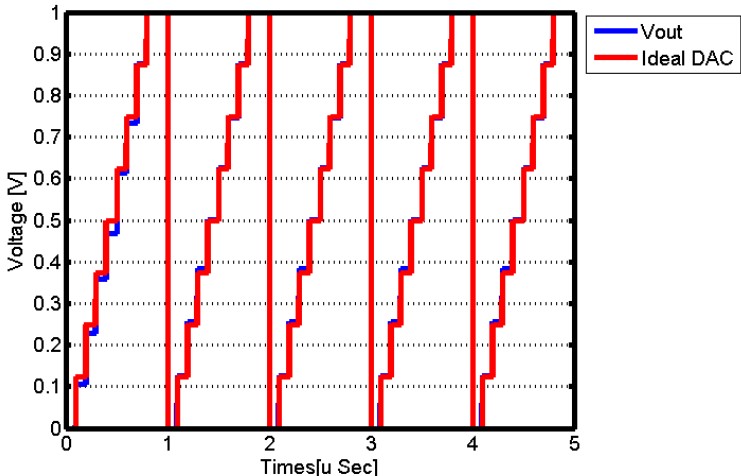

**Figure 10.** Simulation of an eight level input memristor-based DAC circuit.

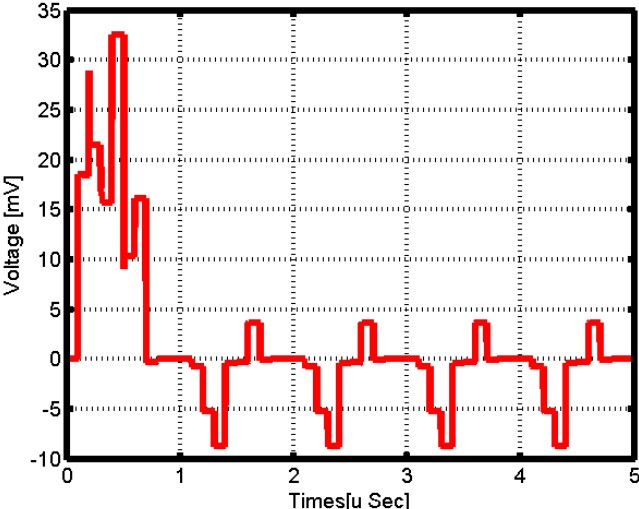

**Figure 11.** Integral non-linearity (INL) simulation for an eight level input memristor-based DAC circuit.

### 4.3. Binary-Weighted DAC Cell-Based on Memristor

Plenty of binary-weighted DAC architectures are used to convert digital word bits to analog signals. Some DAC are quite simple, using few switches and resistors. Other DAC architectures depend on the capacitor network or the source of the transistor current. However, these DAC architectures have limitations in terms of area, speed, resolution, and mismatch error. This section explains the modification that has been applied to the proposed thermometer DAC cell in the previous section in order for it to be used as a binary DAC cell.

In the thermometer DAC cell, it is assumed that the $R_m$ (which is $R_{on}$ or $R_{off}$ according to the state of the memristor) of both memristors is equal. However, in the binary-weighted DAC cell, $R_{m1}$ is double the value of $R_{m2}$, as shown in Figure 12a. Where $D_0$ is a less significant bit, $D_1$ is most significant bit, and $V_{out}$ is the output voltage. Figure 13 shows that the simulation of the 2 bit binary DAC within input voltage for both D0 and D1 are 1 V, and the output voltage varies from 0 V to 1 V. Each step level is 0.33 V. The input frequency of the binary data is 1 MHz.

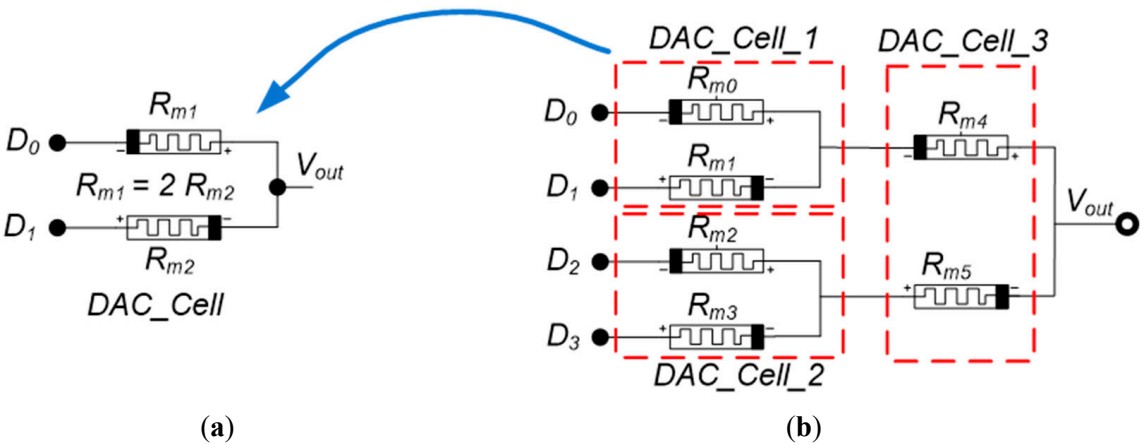

**(a)**            **(b)**

**Figure 12.** A binary-weighted DAC cell. (**a**) A 2 bit DAC cell circuit. (**b**) A 4 bit DAC cell circuit.

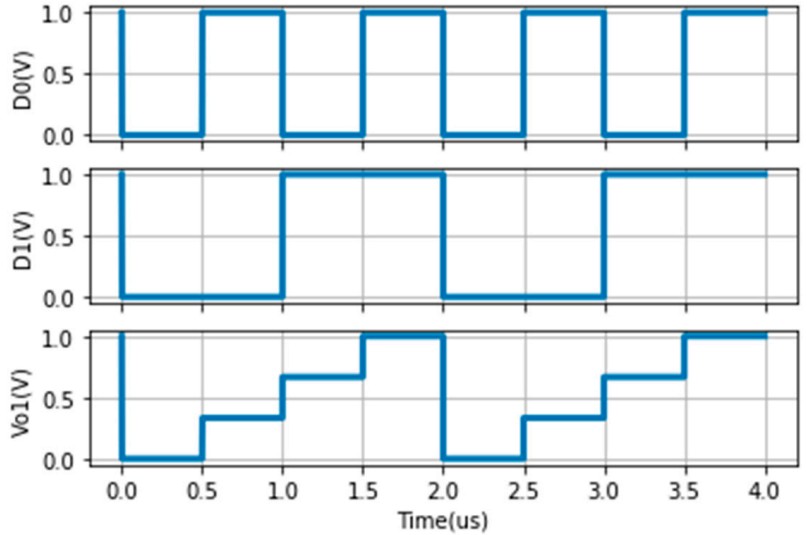

**Figure 13.** Simulation of a 2 bit binary-weighted DAC cell.

Four bit binary data have been constructed, as shown in Figure 12b, where the values of $R_{m0}$, $R_{m2}$, and $R_{m3}$ are double those of $R_{m1}$, $R_{m3}$, and $R_{m5}$, respectively. Figure 14 shows the simulation of 4 bit binary DAC compared with the ideal DAC to study the INL, as shown in Figure 15. The voltage step for 4 bit DAC is 10.5 mV, while the maximum INL

is 3 mV, which is less than the 50% the value of LSB. During analysis of a 4 bit DAC based on memristor technology, a sneak path problem was noticed, which will be explained in the following section.

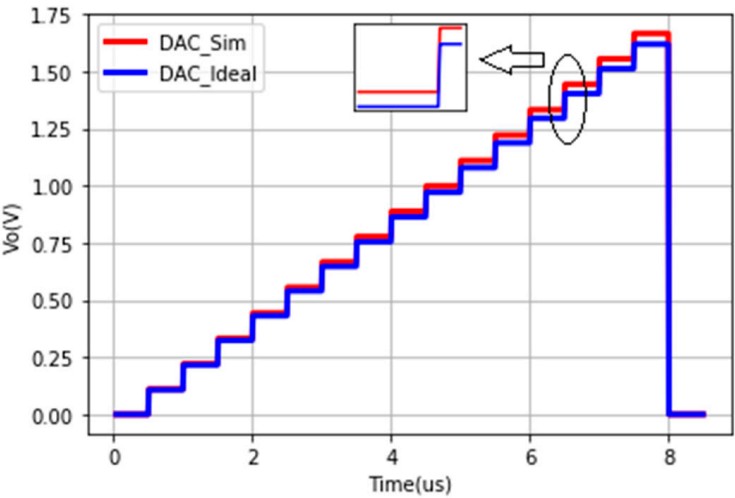

**Figure 14.** Simulation of a 4 bit binary-weighted DAC cell.

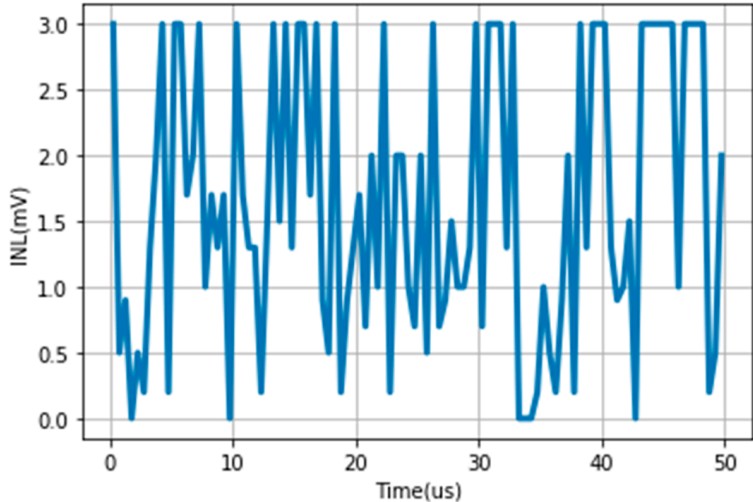

**Figure 15.** INL simulation for a 4 bit input memristor-based binary-weighted DAC circuit.

*4.4. Sneak Path Problem in a Binary-Weighted DAC-Based Memristor*

One of the crucial obstacles for an efficient memristor crossbar array and a cascaded memristor is the so-called sneak path current problem. Some general reviews [21–25] dealing with the fundamental mechanisms, materials, and architectures of memristors have partially addressed the sneak path current problem and its solutions. Figure 16a shows the crossbar array that includes sneak path issues, whereas Figure 16b shows the equivalent circuit for this issue. Generally speaking, there are several recent types of research that have proposed many solutions for this issue, such as the one transistor–one memristor model (1T1M) (as shown in Figure 17), the one diode–one memristor model (1D1M) (as shown in Figure 18), and the one selector–one memristor model (1S1M). In our proposed approach, the sneak path can occur when these cells connected together in a cascade to achieve DAC.

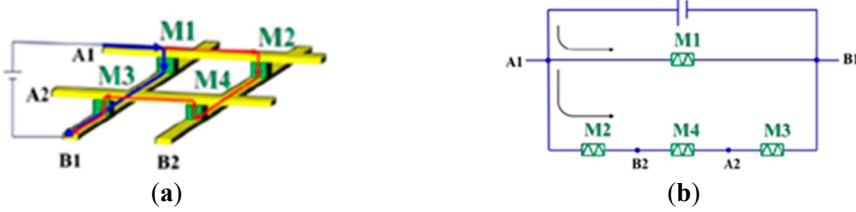

**Figure 16.** Memristors structure and equivalent circuit (**a**) A memristor crossbar array. (**b**) The sneak path problem in circuit.

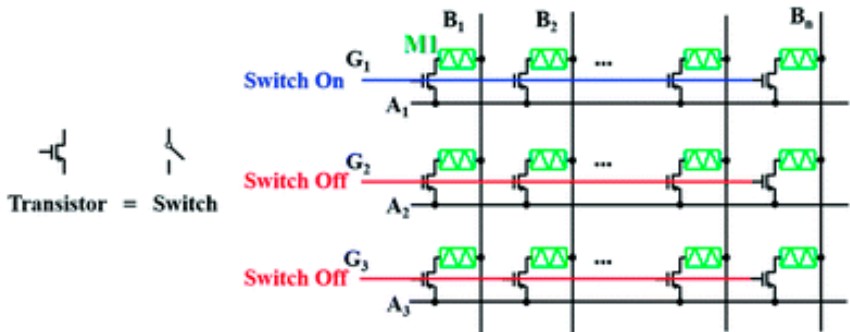

**Figure 17.** A schematic diagram of a one transistor–one memristor (1T1M) crossbar.

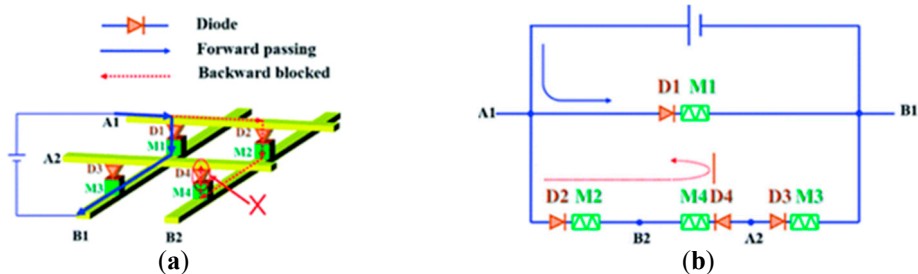

**Figure 18.** Memristors structure and equivalent circuit with 1D1M solution (**a**) A structure diagram of a one diode–one memristor (1D1M) model and (**b**) the equivalent circuit 1D1M solution.

In order to study a sneak path problem (as illustrated in Figure 19), the following binary pattern (1001) is applied. It was noticed that the current passes from $R_{m0}$ to $R_{m1}$ and from $R_{m3}$ to $R_{m2}$ (which are the right direction of the current) affect the functionality of the DAC. However, the most critical path (as shown in Figure 19a) is the red path. In order to prevent the sneak path from occurring, a switch has been added between each cell of DAC, as illustrated in Figure 19b.

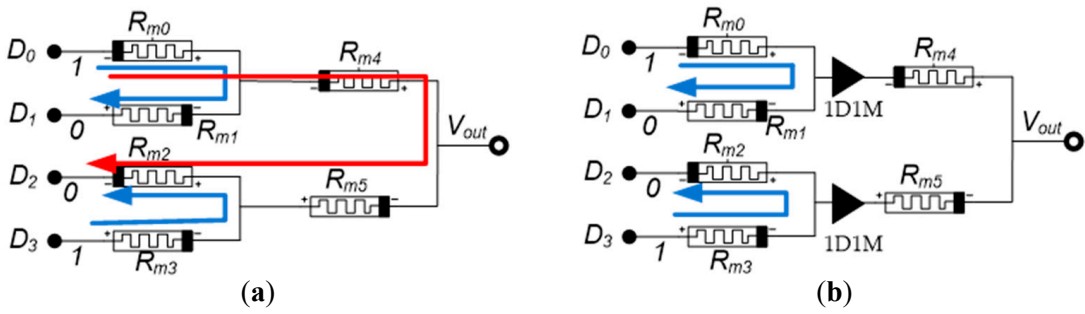

**Figure 19.** The sneak path issue and its solution in a memristor DAC. (**a**) A memristor DAC with a sneak path problem; (**b**) A memristor DAC using the sneak path solution (1D1M).

## 5. Memristor DAC-Based Successive Approximation Register (SAR)-ADC

Successive approximation register (SAR) is one of several analog to digital converters (ADCs). It is suitable for medium-to-high-resolution applications and provides low power dissipation. It requires several comparison cycles to complete one conversion, and it therefore has limited operational speed. SAR architectures are extensively used in low power and low speed applications. Most of the power dissipation is due to the digital control circuit, the capacitive reference DAC network, and the comparator. In recent CMOS technology, the digital control circuits consume less power [26]. In SAR-ADC, there is no component that consumes static power (except for the pre-amplifier, which can be replaced by charging the steering amplifier to cancel this type of power dissipation) [27]. The comparator and capacitor network are the SAR-ADC components that consume the most dynamic power. Several methods have been proposed in recent research to tackle this issue by reducing the switching energy by more than 50%. Figure 20 shows the proposed memristor-based successive approximation register. The capacitor-switching network has been replaced by a memristor DAC network to reduce the area and the switching energy. Figure 21 shows the memristor DAC deployed in the proposed SAR-ADC. This proposed design also deployed a sneak path solution.

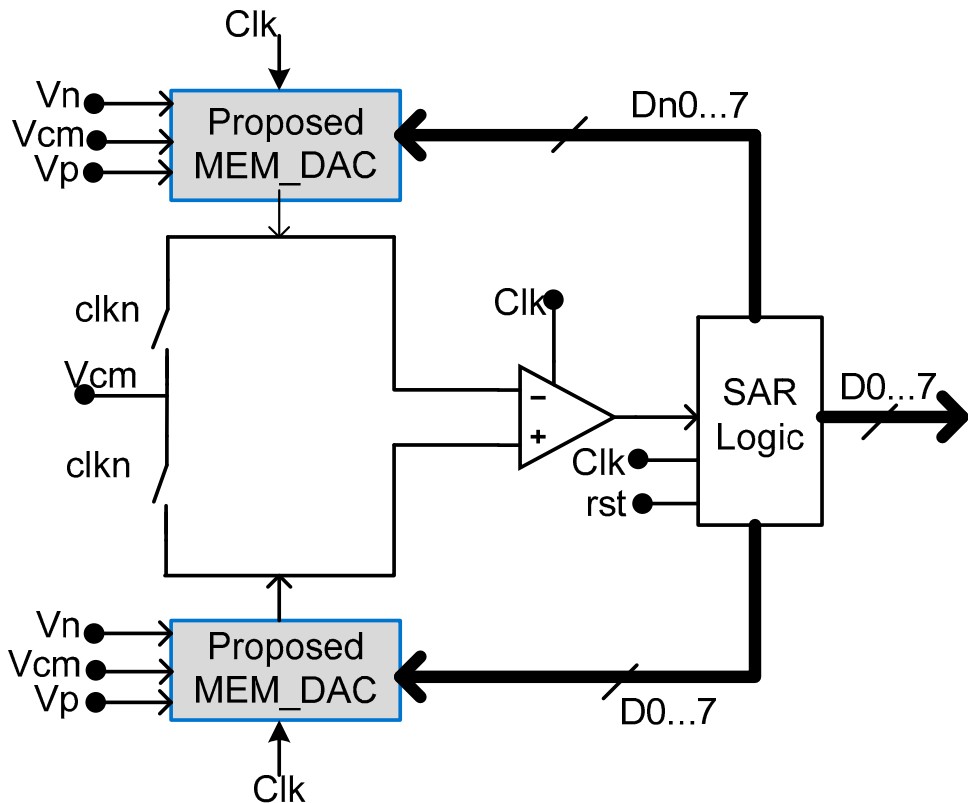

**Figure 20.** The proposed successive approximation register (SAR)-analog to digital converter (ADC) using memeristor DAC.

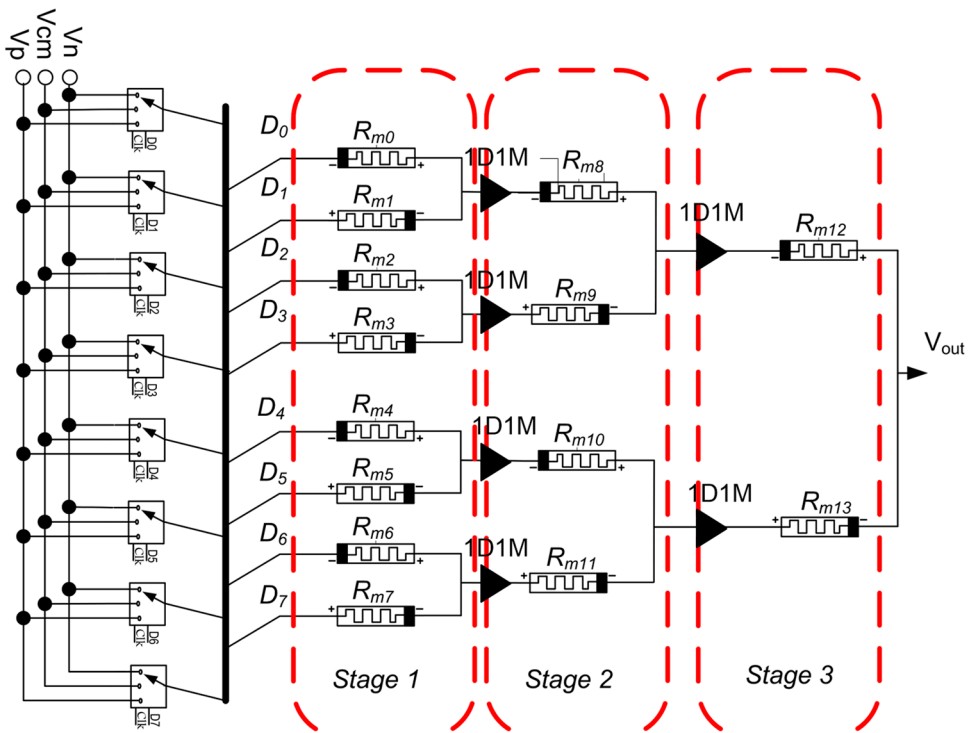

**Figure 21.** Proposed memristor DAC for the successive approximation register (SAR)-ADC.

*Performance of the Proposed SAR-ADC Memristor-Based Circuit*

The proposed design has been simulated using a 65 nm CMOS process and a memristor model using the cadence EDA tool. The input signal for ADC was selected to be 2 kHz, with an input range of 1.2 V peak to peak, a common mode voltage of 0.6V, and a sampling frequency of 1 MHz. To compare the proposed design performance with other ADCs, we used a figure of merit (FoM) expression as follows:

$$FoM = \frac{Power}{F_s.2^{ENOB}} \tag{7}$$

where $F_s$ is the sampling frequencyand ENOB is the effective number of bits (which is $(\frac{SNDR-1.76}{6.02})$). This figure of merit focuses on three vital metrics in ADC: power, $F_s$, and ENOB, with the lowest FoM signaling a better performance. Figure 22 shows the fast Fourier transform FFT of the proposed SAR-ADC model that achieves a signal to noise and distortion ratio (SNDR) of 49.3 dB and a spurious free dynamic range (SFDR) of 61 dB. Table 4 shows the performance of ADC using a power supply of 1.2 V. This design has dissipated by 21 μW, which is a very low level of power consumption. Moreover, it achieves a low figure of merit (87.9 fJ/Conv.-step). The power consumption is, significantly, very low due to the low power consumption of memristors.

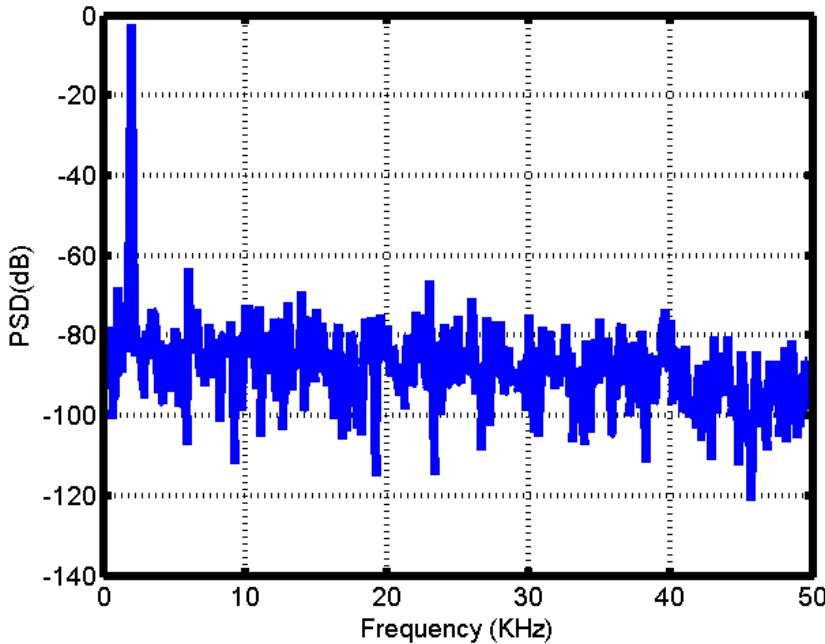

**Figure 22.** The fast Fourier transform (FFT) of the proposed SAR-ADC.

**Table 4.** Performance summary and comparison with state-of-the-art SAR-ADCs.

| Reference | JSSC'11 [28] | VSLI'13 [29] | CASI'16 [30] | IEEEAccsss'20 [31] | IEEETrans.'15 [32] | This Work |
|---|---|---|---|---|---|---|
| Technology | 130 nm | 90 nm | 180 nm | 90 nm | 130 nm | 65 nm |
| Supply Voltage (V) | 1.2 | 1 | 1.8 | 1.2 | 1.2 | 1.2 |
| Sampling Frequency (MHz) | 40 | 30 | 10 | 50 | 0.072 | 1 |
| SNDR (dB) | 50.6 | 56.8 | 66.9 | 57.6 | 82.9 | 49.3 |
| SFDR (dB) | — | 68.6 | 75.8 | 68.8 | 96.8 | 61 |
| ENOB (bit) | 8.26 | 9.16 | 10.82 | 9.26 | 13.5 | 7.9 |
| Power (uW) | 550 | 980 | 820 | 664 | 130 | 21 |
| FoM (fJ/Conv.-step) | 50 | 57 | 44.2 | 21.68 | 156 | 89 |

## 6. Proposed Memristor DAC Comparison

There are few papers have been published on the binary-weighted memristor DAC. Consequently, the parameter for comparison among these proposed memristor DACs is area, in terms of the number of memristors used to achieve the specific DAC. Most of this research focuses on the binary-weighted resistor design technique (which requires a larger number of memristor elements to achieve; in other words, it is difficult to extend to high-resolution DAC due to the limitation of the memristor's properties in implementing a larger memristor element). In this work, the proposed memristor binary-weighted DAC reduced the memristor element required, as shown in Table 5. Moreover, it can be extended to high-resolution DAC without being limited by the memristor's properties. It is clear from Table 5 that the proposed DAC in [5] and [6] used a larger number of memristors and is limited in terms of extended resolutions, while the proposed DAC in [13] has the flexibility to increase DAC resolution. However, it required more transistors (at least one transistor for each extra bit (since the transistor area is larger than the memristor area)). Consequently, the proposed DAC has reduced the area by 40%, as shown in Table 5. Moreover, it can be extended to the n bit of DAC.

**Table 5.** Comparison between the proposed binary-weighted memristor DAC and the results of recent research.

| Item | IEEE 2017 [5] | IEEE 2013/ACM [6] | IEEE 2016/ICMM [9] | This Work | |
|---|---|---|---|---|---|
| Topology | Binary-weighted memristor | Binary-weighted memristor | Current steering memristor | Cascaded memristor | Cascaded memristor |
| DAC Resolutions | 4 bit | 6 bit | 2 bit | 4 bit | 8 bit |
| Number of memristors used | 15 | 63 | 2 * | 9 | 21 |
| Flexibility to increase DAC Resolution | Limited | Limited | Flexible | Flexible | Flexible |

* Two memristors have been used alongside five transistors, one resistor, and one buffer.

## 7. Conclusions

In this paper, 4 bit and 8 bit memristor DACs were proposed using two memristor-based DAC cells. The VTEAM memristor model was employed (with optimized parameters) to simulate the proposed DAC. The sneak path problem in 4 bit and 8 bit DAC has been resolved through use of the 1D1M solution. The proposed DAC achieved 3 mV, which is less than 50% of LSB and the area was reduced by 40% as compared to recent memristor DAC architectures. Moreover, a SAR-ADC model was designed based on the proposed 8 bit memristor DAC. It achieved SNDR 49.3 dB and SFDR 61 dB, with a power supply of 1.2 V and a dissipation of 21 μW. The SAR-ADC realized a figure of merit (FoM) of 87.9 fJ/Conv.-step. The comparison between proposed 4 bit and 8 bit DAC and others memristor DAC [5,6,9] showed that highlighted low area and high resolution were achieved.

**Author Contributions:** All authors have equally participated in contributions to this research article, either in terms of the concept, methodology or validation phases of the proposed protocol. All authors have read and agreed to the published version of the manuscript.

**Funding:** This research received no external funding.

**Conflicts of Interest:** The authors declare no conflict of interest.

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
