# Peer review of "Design of a Memristor-Based Digital to Analog Converter (DAC)"

_electronics, doi:10.3390/electronics10050622_

Round 1

Reviewer 1 Report

Comments:

This work presents an idea of digital to analog converter (DAC) based on memristor. The idea may be novel, but the writing is not suitable for a journal paper. Current version is very hard to understand and ready through. It should be intensively improved before submission. Current decision is “Reject”.

  1. English editing and formatting should be carefully checked by a native speaker or professional editor. Current version is very hard to understand and read fluently.
  2. Abstract is not clear enough for the contribution and novelty.
  3. The abbreviation of some words in the abstract are suggested to provide in the first showing such as VTEAM. Brackets should be written for the abbreviation.
  4. Introduction should be improved to an easy understandable level for common readers. For example, some applications and historical review should be provided before a detail part from some paper. What is the importance or presentative meaning of the selected papers?
  5. Figure 1 from other literature should not be directly presented in this work.
  6. All figures are suggested to re draw for a better understanding.
  7. The comparison between other literatures is not fair since different technology node makes different performance. Authors are suggested to provided the simulation results of conventional DAC to this new memristor-based DAC in the same technology.

Reviewer 2 Report

The paper presents a digital to analog converter (DAC) based on a memristor. The performance was verified by comparing the performance SAR analog to digital converter (ADC).

1. The obvious novelty of this paper should be mentioned except for using the memristor-based DAC because the concept of the memristor-based DAC is already reported in other papers. If the novelty is not enough, it needs more analysis or good performance.

2. Following the performance in Table3, the ADC performance is not good to compare to other works. Although it was mentioned the limitation of FoM is due to the low sampling frequency, in general, it is more difficult to achieve good FoM in the case of the high sampling frequency. (In General, the FoM of ADCs which operates at low sampling frequency is more good.)

3. The quality of some figures is not good. It should be redrawn by considering the unity.

4. I still doubt how the memristor ON/OFF resistance can be made consistent values in a real case (not a simulation case). If the memristor has the mismatch, how change the ADC performance? 

In case compare CAP DAC, what is a obvious advantage? 

Reviewer 3 Report

Paper presents design of memristor based DAC of two types: thermometer DAC and binary-weighted DAC with case studies. Some issues should be addressed before considering this paper for publishing:

-Comparison of proposed DAC design with similar DAC memristors available in literature should be included.

-Fig. 1 Do authors have permission to used the following image? Please provide proof of the permission or use another image.

-Reference are not citated where required, e.g., Section2 first paragraph (lines 91-98) when describing models’ appropriate reference should be included.

- Section 2.1. please clarify VTEAM model parameters and simulation setup used for simulation as you stated in Cadence tool.

- Grammar errors and syntax error occur in many places in the text. Native English speaking lector should proofread the manuscript.  

- Author claims low sampling frequency of proposed solution, please comment on possible methods to improve FoM and sampling frequency.

- In reference section all the references should be citated in the same manner which is not the case now

Round 2

Reviewer 1 Report

Although the quality of this manuscript had been improved slightly. The format and English are not accepted to be published. Response of Q1 in previous run didn't show enough quality. Current status of this work is "Major revision". Authors are suggested to provide a clearly and fully response to each question, not only few sentences. The abstract writing is still not friendly to common readers. The introduction is not acceptable from a reference in the beginning, which can be too narrow for a systematic study. The quality of comparison table is poor. Authors are suggested to put more efforts to improve overall quality, not just a minimum improvement.

Author Response

We would like to thank the reviewer for his positive review and the valuable remarks that returned to us in order to improve our work.

The revised version has been attached in two highlighted colors.

Yellow is the first modification. 

Blue is the second modification.  

Reviewer 2 Report

No comments

Author Response

(The authors gave the same response as above.)

Round 3

Reviewer 1 Report

The quality of manuscript had been improved to be published. However the English editing is strongly suggested before publish.